# Moving the Needle Forward in Genomically-Guided Precision Radiation Treatment

**DOI:** 10.3390/cancers15225314

**Published:** 2023-11-07

**Authors:** Andrew Tam, Benjamin D. Mercier, Reeny M. Thomas, Eemon Tizpa, Irene G. Wong, Juncong Shi, Rishabh Garg, Heather Hampel, Stacy W. Gray, Terence Williams, Jose G. Bazan, Yun R. Li

**Affiliations:** 1Department of Radiation Oncology, City of Hope Comprehensive Cancer Center, 1500 E Duarte Rd., Duarte, CA 91010, USA; atam@coh.org (A.T.); bmercier@coh.org (B.D.M.); rthomas7@sgu.edu (R.M.T.); etizpa@coh.org (E.T.); igwong@caltech.edu (I.G.W.); jshi@coh.org (J.S.); rgarg@coh.org (R.G.); terwilliams@coh.org (T.W.); 2Department of Medical Oncology & Therapeutics Research, City of Hope Comprehensive Cancer Center, 1500 E Duarte Rd., Duarte, CA 91010, USA; hhampel@coh.org (H.H.); stagray@coh.org (S.W.G.); 3Department of Cancer Genetics and Epigenetics, City of Hope National Medical Center, Duarte, CA 91010, USA; 4Division of Quantitative Medicine & Systems Biology, Translational Genomics Research Institute, 445 N. Fifth Street, Phoenix, AZ 85022, USA

**Keywords:** radiation sensitivity, radiogenomics, genes, germline, DNA, radiation treatment, radiation treatment, oncology, cancer, mutations

## Abstract

**Simple Summary:**

Genetic information is seldom incorporated in formulating radiation treatment recommendations for patients with cancer, even though genetic information is now well established to be prognostic and predictive of cancer outcomes and response to systemic therapy. With the increasing accessibility to and use of genetic testing, tumor, and germline genetic data have the potential to inform clinical decisions by improving the efficacy of radiation treatment and ensuring the safety of treatment delivery. This review summarizes the biological underpinning of “radio-sensitizing genes”, discusses the clinical basis and evidence for identifying genetic mutations as radiation response biomarkers, and proposes future directions for research as well as clinical implementation.

**Abstract:**

Radiation treatment (RT) is a mainstay treatment for many types of cancer. Recommendations for RT and the radiation plan are individualized to each patient, taking into consideration the patient’s tumor pathology, staging, anatomy, and other clinical characteristics. Information on germline mutations and somatic tumor mutations is at present rarely used to guide specific clinical decisions in RT. Many genes, such as *ATM,* and *BRCA1*/*2*, have been identified in the laboratory to confer radiation sensitivity. However, our understanding of the clinical significance of mutations in these genes remains limited and, as individual mutations in such genes can be rare, their impact on tumor response and toxicity remains unclear. Current guidelines, including those from the National Comprehensive Cancer Network (NCCN), provide limited guidance on how genetic results should be integrated into RT recommendations. With an increasing understanding of the molecular underpinning of radiation response, genomically-guided RT can inform decisions surrounding RT dose, volume, concurrent therapies, and even omission to further improve oncologic outcomes and reduce risks of toxicities. Here, we review existing evidence from laboratory, pre-clinical, and clinical studies with regard to how genetic alterations may affect radiosensitivity. We also summarize recent data from clinical trials and explore potential future directions to utilize genetic data to support clinical decision-making in developing a pathway toward personalized RT.

## 1. Introduction

In the last decade, the increased availability and rapid integration of genetic testing have revolutionized the landscape of cancer care, particularly in the management of patients with advanced and metastatic malignancies. It has paved the way for the development of personalized treatment recommendations by utilizing mutation profiles to target specific underlying molecular drivers of a tumor [1]. Despite the progress that has been made in our understanding of cancer genetics, the field of radiation oncology has lagged behind medical oncology in incorporating specific genetic information in formulating our decisions surrounding radiation treatment (RT). Even though the field has dramatically advanced in treatment delivery and planning techniques, RT is still largely prescribed today as it has been carried out for many decades, based on tumor size, disease type, histology, surgical margin status, disease stage, and proximity to normal anatomical structures [1,2].

It has long been observed that individuals with certain genetic disorders are predisposed to adverse side effects from RT. One of the most well-known examples is ataxia-telangiectasia (AT), which is a condition caused by homozygous or complete loss of function mutations in the *ATM* (ataxia telangiectasia, mutated) gene leading to impaired response to DNA (deoxyribonucleic acid) double-strand breaks [3]. Reports from as early as the 1960s have documented the radiation sensitivity of patients with AT developing more pronounced toxicities and increased risk of secondary malignancy from radiation than the general population [4,5]. Since then, many more mutations associated with radiation sensitivity have been identified and the list of potential “radio-sensitizing genes” continues to grow and is likely only to expand exponentially in the next decade.

Perhaps more importantly, radiation oncologists will be increasingly faced with clinical scenarios in which patients will present with existing genetic testing results. The radiation oncologist must then decide based on this information, whether specific mutations portend increased risks of treatment toxicity or resistance, whether radiation would be recommended, and whether adaptations in the treatment field or dose need to be made based on mutations in “radio-sensitizing genes”. Consensus on RT recommendations for patients with pathogenic variants, even those with mutations in genes shown in the laboratory to affect radiation response, such as *ATM*, is lacking. Clinical guidelines are needed to guide radiation oncologists on how to safely incorporate genetic information in treatment decision-making for patients with “radio-sensitizing” mutations.

The field of radio-genomics, herein defined as the study of genetic/molecular variation associated with response to therapeutic radiation as it relates to tumor control and normal tissue toxicity, is still in its infancy. In this review, we aim to summarize and examine existing data from both bench and clinical research to provide an updated review of the role of genetic alterations on radiation response. This summary is intended to provide practicing radiation oncologists, radiation biologists, and cancer genetics researchers an overview of the genetic basis of radiosensitivity, help inform future research efforts, and emphasize the need for the creation of clinical guidelines surrounding how genetic information should be incorporated in clinical radiation oncology treatment decision-making.

## 2. Genetic Basis of Radiation Sensitivity

Radiation destroys cancer cells by damaging genomic DNA via direct DNA breakage or indirectly via the formation of free radicals and other reactive oxygen species (ROS) [6]. Radiation consequentially induces DNA base damage, single-strand breaks (SSBs), and double-strand breaks (DSBs) to the DNA [6]. Thus, “radiation sensitivity” (or “radio-sensitivity”) describes any toxic, cancerous, or aging effect resulting from radiation [7]. In some of the existing literature, the stochastic effect of radiation in inducing secondary cancer is included in the definition of radiation sensitivity [7], however, in this review, we will primarily focus on the interplay between genetic mutations and their roles in affecting clinical treatment response.

Radiosensitivity can be subcategorized to refer to the responsiveness of tumor cells or the reaction of normal tissues to radiation. Radiosensitivity of tumor cells can enhance cell killing and thereby improve the efficacy of radiation, but conversely, radiosensitivity of healthy cells can lead to more prominent radiation-induced side effects. The molecular underpinning conferring radiosensitivity is complex and not fully understood. However, there is mounting evidence supporting the importance of genetic alterations in affecting clinical radiosensitivity. It has long been observed that patients with certain genetic syndromes develop adverse, sometimes, fatal toxicity following therapeutic radiation. In addition to the example of AT as mentioned above, cancer patients with Fanconi’s Anemia, a rare autosomal recessive DNA repair disorder, who received RT have been shown to exhibit hypersensitivity to radiation [8,9,10,11]. Table 1 summarizes the known genes, from both pre-clinical and clinical studies, that have been identified as being correlated with increased toxicities from radiation.

In brief, genomic characteristics that influence radiosensitivity are as follows [6,7,12] (Figure 1): Cell cycle. Cells that are at or close to the mitotic (M) phase are the most sensitive.Cell cycle checkpoint. Mutated cells that lack checkpoint function proceed to the M phase with damaged chromosomes leading to a higher risk of cell death.DNA repair. Improper DNA repair functioning leads to the maintenance of genetic instability or misrepair of DSBs causing cell lethality.

As shown in Table 1, most of the “radio-sensitizing genes” have critical roles in the cell cycle checkpoint or DNA repair pathways (#2 and #3 of the abovementioned characteristics). Mutations of genes in these pathways can lead to improper or reduced functioning of cellular machinery in response to radiation-induced damage. This, in turn, leads to cell death and, if the amount of tissue damage is significant, results in clinically observed toxicities.

**Table 1 cancers-15-05314-t001:** Genes Associated with Radiosensitivity.

Gene Symbol	Syndrome	Clinical References	In Vitro/Preclinical References	Level of Evidence
*ATM*	Ataxia-telangiectasia	[9,13,14,15,16,17,18,19,20]	[21,22,23,24,25,26,27,28,29,30]	****
*ATR*	Seckel syndrome	n/a	[31]	*
*BLM*	Bloom syndrome	[32]	[33,34,35]	**
*BRCA1/2*	n/a	[13,36]	[37,38,39,40,41]	****
*CHK1/2*	n/a	n/a	[42,43,44,45,46,47,48,49,50]	**
*FA* complementation groups, including *FANCA*	Fanconi anemia	[8,51,52,53]	[35,54,55,56,57,58]	****
*LIG4*	Ligase IV syndrome	[59]	[30,60,61,62]	***
*MLH1, MSH2*	Lynch syndrome	n/a	[63,64]	*
*MRE11*	Ataxia-telangiectasia-like disorder	n/a	[65,66]	*
*NBS1/NBN*	Nijmegen breakage syndrome	[67,68]	[29,69,70,71,72,73,74]	***
*NF1*	Neurofibromatosis type 1	n/a	[75,76]	*
*PARP1*	n/a	n/a	[77,78,79,80,81]	**
*PTEN*	Cowden syndrome	n/a	[82,83,84,85]	**
*PTCH1*	Gorlin syndrome (nevoid basal cell carcinoma syndrome)	n/a	[86,87,88]	**
*RAD50*	Nijmegen breakage-like syndrome	n/a	[89,90]	*
*RAD51*	n/a	[91]	[92,93,94,95,96,97]	***
*RB1*	Retinoblastoma	n/a	[98,99,100]	**
*RNF168*	RIDDLE syndrome	n/a	[101,102,103]	**
*SMC1A*	Cornelia de Lange syndrome	n/a	[104]	*
*STAT3*	Hyperimmunoglobulin E syndrome	n/a	[105]	*
*TP53*	Li-Fraumeni syndrome	[106]	[27,28,107,108]	***
*XPD/ERCC2*	Xeroderma pigmentosum	n/a	[109]	*
*WEE1*	n/a	n/a	[110,111,112,113,114]	**

Assessment of the level of evidence is based on the quality and amount of currently available studies supporting the role of each gene in conferring radiosensitivity using the following scale: **** strong evidence, with several clinical and preclinical studies supporting the role of the gene; *** high-moderate evidence, with only a few clinical studies, but several preclinical studies; ** low-moderate evidence, with no clinical studies, but several preclinical studies; * low evidence, with only a few preclinical studies.

However, the correlation between the presence of deleterious mutations and radiation sensitivity is far from direct. This is due to inherent individual heterogeneity and the complex biology of the function of encoded proteins in often redundant biological pathways. Moreover, many other factors may affect the impact of a given mutation, such as the extent of the mutation’s impact on expression or function, i.e., penetrance, whether the mutation acts in a dominant or recessive manner, dose effects and compensatory biological mechanisms, alternative pathways and/or redundant proteins. In the case of the vast majority of mutations, there are alternative biological mechanisms to ensure sufficient genomic DNA integrity prior to cell division. These compensatory mechanisms are complex with regulatory means at multiple levels, such as [115]:Genomic level (such as mutation or chromosomal aberration)Transcriptome level (such as mRNA [messenger ribonucleic acid] expression)Epigenetic level (such as DNA methylation)

Additionally, for each gene, the physical location of the mutation, as well as its functional consequence, can significantly affect whether a particular mutation is deleterious. For example, a given single nucleotide polymorphism (SNP) in the coding DNA may cause missense, non-sense, or silent mutations at the protein level, which can further increase the complexity of interpreting mutation data. Multiple early Genome-Wide Association Studies (GWAS) on identifying SNPs were some of the first efforts in understanding the genomic-level alterations in association with radiation toxicity. SNPs of the *TGFb1* gene were one of the first gene polymorphisms found to be associated with radiotoxicity in prostate cancer and rectal cancer patients [116,117]. Other notable SNPs include *XRCC1*, rs2682585 which were found to be significantly associated with lower risk of skin toxicities and overall toxicity in a cohort of 753 breast cancer patients [118], and four SNPs (rs17055178, rs10969913, rs11122573, and rs147121532; gene functional studies revealed that these variants lay in gene regulatory regions of the genome) associated with increased urinary frequency, hematuria, decreased urinary stream, and rectal bleeding in prostate cancer patients of European ancestry [119].

While some early GWAS studies have shed light on possible pathogenetic pathways of radiosensitivity, others have resulted in dead ends or conflicting evidence regarding their clinical utility. Ongoing research effort aims to study the functions of specific DNA mutations or combinations thereof, yet many have yet to be determined and consequently, the majority of mutations identified in patients are reported as “variants of unknown significance” (VUS), even when they occur in established radiation sensitivity genes such as *BRCA* or *ATM*. Beyond SNPs, other types of mutations such as insertion-deletions, large-scale translocations, and copy number variations are even more complex to functionally interpret, particularly those mapping to non-coding, but potentially regulatory regions. Below, we will review the functioning of a few selected genes in radiation response to discuss the implications of these genes in radiosensitivity.

## 3. Gatekeeper and Caretaker Genes

### 3.1. Cell Cycle Checkpoint “Gatekeeper” Genes

Genes that regulate the cell cycle have critical roles in ensuring the DNA integrity of the cell prior to proceeding with cell division. Hence, they are often referred to as “gatekeeper” genes. The cell cycle is a series of cellular events that facilitate the replication of DNA and cell division to create two daughter cells [120]. The normal function of these checkpoint genes is to detect aberration in the DNA and control the progression of the cell cycle. Two examples of classical gatekeeper genes are *p53* and *ATM*. Given their functions in mediating response to DNA damage, these genes have long been considered key players in radiosensitivity.

When DNA damage is detected, these gatekeeper genes either arrest the cell cycle to allow for DNA repair or induce cell death [120]. Checkpoints in the cell cycle include the Gap 1 (G1) and DNA Synthesis (S) phase junction (G1/S) to ensure to prevent replication of damaged DNA and the Gap 2 and mitosis (G2/M) for prevention of segregation of aberrant chromosomes into daughter cells [120].

*Ataxia-telangiectasia, mutated* (*ATM* gene) is the master controller for both “gatekeeper” and “caretaker” signaling pathways. In response to DNA damage, the *ATM gene product,* ATM, exhibits protein kinase activity and phosphorylates p53 at the G1-S checkpoint to induce cell cycle arrest and promote DNA repair [3,20]. Carriers of a germline mutation in *ATM* are notably at an increased risk of radiation sensitivity [3,20]. A study on female breast cancer patients revealed a significant correlation between the presence of *ATM* mutation and the development of Grade 3–4 toxicity, such as fibrosis (*p* = 0.001) following RT [17]. More recently, a meta-analysis on 5456 breast and prostate cancer patients found patients who are carriers with *ATM* rs1801516 SNP reported odds ratios of approximately 1.5 for acute and 1.2 for late toxicity after RT [20].

As mentioned above, p53 is one of the main downstream effectors of ATM. p53 is a gene product of the tumor suppressor gene *TP53* and it signals multiple downstream targets in response to DNA damage that ultimately determines cell fate after signature radiation exposure [121]. Patients with germline *TP53* mutations (for which single copy loss of function is the most common cause of Li-Fraumeni Syndrome) are predisposed to develop early-onset cancers of multiple tissue types [122]. However, the role of the *TP53* mutations in radiosensitivity is not as fully understood as p53 has innumerable roles in normal cell physiology. O’Connor and colleagues studied the effect of ionizing gamma-radiation in 60 different cell lines, 39 of which contained a variant mutation of *TP53* [108]. Cells possessing mutant variants of *TP53* showed a characteristic inability to induce cell cycle arrest in G1 despite a high DNA damage burden. As a result, cell growth remained uninhibited compared to cells possessing wildtype *TP53*. While O’Connor and colleagues were unable to differentiate and characterize the various mutations present in *TP53* for each mutant cell line, since then many specific mutations such as *TP53* A135V have been established to increase radio-resistance. *TP53* knockout cells demonstrated a remarkable decrease in cell viability when exposed to gamma-radiation compared to their wildtype-*TP53* counterparts [123].

The Retinoblastoma tumor suppression gene (*RB*) is another cell cycle checkpoint gene heavily involved in the mediation of the G1/S checkpoint [100]. The products of this gene are integral to the antiproliferative process and have been characterized to be aberrant or even abrogate in approximately 20–35% of breast cancers as well as occupying a very prominent role in the carcinogenesis of many human lung adenocarcinomas [100,124]. *RB* is also known to act as an effector in the MAPK signal amplification pathway as well as possess an integral role in the formation of the DREAM complex alongside *LIN37* [124,125]. The DREAM-p53 pathway is vital for the function of the G1/S checkpoint [125]. Breast cancer cells characterized as deficient in *RB* expression were found to be more sensitive to both radiation and cisplatin when compared to wildtype donor cells [100]. However, there is scant literature available affirming radiosensitivity in other types of human cancer cells or clinical settings.

*CHK1* and *WEE1* play integral roles in the regulation of the G2/M checkpoint of the cell cycle. In vitro studies of WEE1 and CHK1/CHK2 inhibitors in human colorectal cancer and osteosarcoma cell lines have been shown to increase sensitization to both RT and chemotherapeutic drugs [111,126]. Similarly, there is in vitro evidence suggesting enhanced radiosensitivity with CHK1 inhibition using a selective inhibitor, MK-8776, in human triple-negative breast cancer cell lines [42]. Furthermore, some early phase I clinical data on the addition of WEE1 inhibitor AZD1775 in patients with pancreatic cancer when combined with RT and gemcitabine have suggested improved survival outcomes compared to historical data [127]. While the results of said in vitro and clinical studies show promise, further investigation is needed to validate the evaluation of these genes in precision medicine and clinical practice in radiation oncology.

### 3.2. DNA Repair “Caretaker” Genes

DNA repair (or caretaker) genes often operate in conjunction with gatekeeper genes, and at times with overlapping purposes. Many gatekeeper genes function in the regulation of cell proliferation and cell cycle regulation. DNA repair genes encode proteins directly involved in DNA repair processes in repair mechanisms of both SSBs and DSBs, as well as DNA mismatch repair (MMR). Such repair mechanisms include non-homologous end joining (NHEJ), homologous recombination (HR), base excision repair, and nucleotide excision repair. The resulting encoded proteins from these genes often directly interface with the DNA molecule [128]. However, the line between DNA repair and gatekeeper genes is not always distinct. Gene products from genes such as *BRCA1/2, TP53*, *RAD9*, and *ATM* possess features of both cell cycle checkpoint and DNA repair regulation as well as intrinsic DNA repair features [129]. Mutations in DNA repair genes that cause functional attenuation will likely lead to the accumulation of mutations in other genes, which can contribute to cancer risk [130]. While hundreds of genes are implicated in DNA repair processes, we will discuss those with the strongest evidence as being important in individual sensitivity to radiation.

Lynch syndrome, or hereditary nonpolyposis colorectal cancer, is a well-documented inherited disorder that resulting from a constellation of mutations in MMR genes *MLH1*, *MSH2*, *MSH6*, or *PMS2* [131]. The MMR proteins encoded by these genes act to directly interface with the damaged DNA molecule via heterodimerization with one another [132]. While Lynch syndrome was initially associated with the occurrence of colorectal cancer, with up to 20% of all patients with colon cancer being afflicted by this genetic condition, these patients are also at increased risk of other solid tumors, including endometrial, ovarian, and gastric cancers [131,133]. However, there is limited evidence suggesting increased radiosensitivity among patients with Lynch syndrome. Studies in *MLH1*-deficient murine models suggest that there is an elevated risk of the development of radiation-induced tumorigenesis 72 weeks after birth after exposure to radiation (2 grays [Gy]) at 2-week or 10-week of age [63,134]. An increase in loss of *MLH1* staining has been associated with the onset of therapy-related colorectal cancers in patients previously treated for Hodgkin’s lymphoma. While most studies demonstrated a radio-sensitizing effect associated with MMR deficiency, others found that it conferred radio-resistance in the setting of chemoradiation [134].

*RAD51* is a gene family encoding proteins directly associated with DNA repair, specifically in HR in the setting of DSBs [135]. During HR, RAD51 is involved in the homology search, strand invasion, and strand pairing to facilitate repair [135,136]. *RAD51* and its paralogs are heavily associated with tumorigenesis, as *RAD51* tends to be downregulated in many cancers, causing a notable decline in DNA repair capacity, as well as overexpressed in others [129]. Overexpression of RAD51 is known to contribute to anomalous recombination between both short repetitive sequences and homologous sequencing, resulting in a significantly increased likelihood of tumorigenesis [129]. Additionally, mutations in *RAD51* paralogs are known to cause sensitivity to replicative DNA damage [135]. The level at which *RAD51* and its paralogs are expressed in a given malignancy could be immensely important in determining radiosensitivity. In osteosarcoma and prostate carcinoma cell lines, *RAD51* knockdown was associated with an increase in radiosensitivity [92,93]. Specifically in prostate carcinoma cell lines DU145 and PC-3, EGFR inhibitors were utilized to downregulate the expression of *RAD51*, potentiating the effects of ionizing radiation on cell proliferation [93].

*BRCA1/2* are the most commonly affected genes in hereditary breast and ovarian cancer, though these mutations impact the risk for many other cancers including prostate and pancreatic cancers [137]. Both *BRCA1* and *BRCA2* are involved in the maintenance of HR, a DSB repair pathway, at sites of DNA damage. *BRCA1* encodes the breast cancer type 1 susceptibility protein (BRCA1), a phosphoprotein that helps maintain genomic stability as a component of the multi-subunit protein complex BRCA1-associated genome surveillance complex (BASC) [138]. BASC interacts with RNA polymerase II and histone deacetylase, displaying a role in transcription, repair of DNA double-stranded breaks (DSBs) [138], and recombination [138]. BRCA1 colocalizes with BRCA2 and the recombinase RAD51 to activate RAD51-mediated HR of DSBs by helping RAD51 assemble on ssDNA to search for a homologous DNA repair template and initiate strand exchange [139]. Tumors harboring *BRCA1/2* mutations have been shown to have increased sensitivity to radiation. For example, a study reported a higher objective response rate to RT (*p* = 0.007) and a lower cumulative 1-year local recurrence rate (*p* = 0.008) in solid tumors carrying *BRCA1/2* somatic mutations compared to the *BRCA* wildtype group, indicating *BRCA1/2* loss of function leads to elevated radiosensitivity through blockade of HR [13]. Of note, *BRCA2* may have more of a critical role in radiosensitivity than *BRCA1.* Some studies have suggested that cells with heterozygous germline mutation of *BRCA2* exhibit decreased DSB repair capacity, but not in *BRCA1*, after irradiation [41]. Clinically, patients with either germline heterozygous *BRCA1* or *BRCA2* mutations have not been shown to experience increased radiation sensitivity compared to controls in patients with early-stage breast cancer after breast-conserving RT [140].

DNA repair genes and gatekeeper genes, while differing in terms of functionality, have similarly important roles in conferring the radiosensitivity of a given cell (Figure 2). With the recent advent of routine clinical cancer mutation profiling, a comprehensive assessment of gatekeeper and DNA repair genes can be employed to optimize cancer therapy. In the next section, we will explore ways that radiation genomics can change the practice of Radiation Oncology.

## 4. The Role of Molecular Testing in Identifying Radiosensitizing Genes 

Despite the improved understanding of the molecular “gatekeeper” and “caretaker” pathways in recent years, establishing the role of a particular gene responsible for radiosensitivity remains a challenge, especially in light of potentially conflicting evidence. For example, patients with Fanconi anemia demonstrate radiosensitivity clinically, but with inconsistent findings on the extent of DNA repair in response to radiation from in vitro studies [141]. Another challenge is in identifying new potential genes that lead to radiosensitivity and ongoing efforts have aimed to tackle this by leveraging toxicity outcomes from large clinical datasets and also at the individual patient level.

As mentioned above, GWASs have revolutionized the search for candidate genes via “tagging” hundreds of thousands of SNPs through the use of microarray [142,143]. These studies utilize large patient data to identify potential genetic variants associated with a certain toxicity. The UK RAPPER study was one of the largest studies in identifying potential SNPs associated with chronic toxicities from adjuvant breast RT or definitive prostate RT [144,145]. Another notable GWAS study by Kearns and colleagues combined the RAPPER study cohort with cohorts from RADIOGEN, GEN-PARE, and the Cancer Centre in Canada (CCI) studies identified four SNPs associated with rectal bleeding, decreased urinary stream, and hematuria after prostate cancer RT [119]. The list of concerned SNPs continues to expand from other GWAS studies [116,119,146,147]. However, due to limited validation and concern for a high number of false positives [148], and more notably, some of these initial findings could not be validated in additional cohorts. Thus, few of these findings have been successfully translated into clinical practice [144,149].

While functional bioassays are routinely utilized to understand how specific DNA mutations can affect radiation response, the role of functional bioassays remains limited in clinical settings. For patient care, many other types of molecular tests are employed instead. At the genomic levels, whole exome, whole genome and targeted panel based sequencing employs next-generation sequencing (NGS) technology to interrogate mutations in the genome and subsequent filtering based on quality criteria such as by percentage of reads showing the variant, then by variants outside of coding regions and known variants to identify a potential gene of interest [150]. At the chromosomal level, fluorescence in situ hybridization (FISH) uses fluorescence tags to identify chromosomal aberration, such as translocations, amplifications, and deletions [151]. At the RNA level, RNA sequencing (RNAseq) applies the same principle as NGS to assess the functional levels of genetic variation [152]. Lastly, the use of tests at the protein level, such as immunohistochemistry (IHC), has now been commonplace in our daily oncology practice for example in the detection of HER2 status [153].

## 5. Moving towards Genomically-Guided Radiation Treatment 

The increasing accessibility of genetic tests will undoubtedly accelerate our efforts in detecting mutations in individual patients. As described above, with the mounting evidence supporting the role of genes in radiosensitivity, there is an urgent need to consider the incorporation of genetic information in our approach to RT. By leveraging genetic mutation information, RT can potentially improve its effectiveness by further maximizing tumor cell killing and minimizing toxicities to surrounding normal tissues. However, the challenge remains in how to translate the current evidence into our clinical practice in changing patient outcomes. Here, we explore four areas where genetic results can help guide clinical decisions and the implications for future research in our field.

### 5.1. Omission of Radiation Treatment

As described above, in some individuals with germline mutations in radio-sensitivity genes, radiation may lead to potentially significant clinical toxicities, including death [12]. The current NCCN guidelines advise avoidance of RT when possible among patients with Li-Fraumeni syndrome (i.e., most commonly a result of deleterious at least single copy loss of function mutation in *TP53*) due to concern for an increased risk of secondary malignancy [154,155]. However, there is no such recommendation for any other genetic syndrome or mutations on the basis of radiosensitivity.

With an increasing identification and understanding of the role of radio-sensitizing genes, targeted mutation profiling of these genomic regions can potentially provide valuable insights into the appropriateness of RT for patients who are known carriers. The utmost challenge is how to incorporate genetic information in weighing the risks versus benefits of omitting radiation when radiation is typically clinically indicated and/or is the best treatment option. In situations where patients are found to have predicted deleterious mutations in radio-sensitivity conveying genes and where there are good alternative treatment options that have equivalent patient outcomes, clinicians should carefully consider these instead of radiation. For example, in patients with a potentially deleterious mutation in p53 with prostate cancer, opting for prostatectomy would be a very reasonable alternative in the setting of uncertain but possibly high risk of secondary cancers after definitive radiation. In situations where radiation is the optimal therapeutic modality or provides the only reasonable chance of cure, most patients with inherited cancer susceptibility syndromes should still receive radiation.

While this review has largely focused on single gene mutations, the use of polygenic risk scores to simultaneously consider the role of mutations in multiple radiation response genes may have important future potential to inform the clinical use of genetic data. Such an approach will require heavy investments in future efforts at predictive modeling in real-world and large-scale datasets. The ideal predictive model would utilize information, not solely from a single gene, but from a profile of genetic mutations to assess the potential risk of toxicity from radiation.

### 5.2. Radiation Dose to Tumor

Another consideration is the use of genetic information in guiding radiation dose. Our current practice of RT operates on the assumption that tumors of the same type respond similarly to radiation, regardless of the potential radiosensitivity of the targeted tumor or patients’ normal tissues. This one-size-fits-all approach leads to the same prescription dose across patients. However, this can pose challenges for patients with radiosensitivity, as it may exacerbate normal tissue damage and induce adverse side effects. A noteworthy study by Scott and colleagues addressed this concern by developing a clinical model to optimize RT dosages based on a retrospective cohort of non-small cell lung cancer patients [156]. Their findings revealed that a considerable number of patients were at risk of being overdosed, thereby increasing the likelihood of treatment-related toxicity [156]. Studies such as the one mentioned above further underscore the necessity for personalized RT regimens tailored to the specific characteristics and heterogeneity of individual tumors.

The use of molecular data to guide radiation dosing is still in its infancy. A good recent example is the use of p16 overexpression to reduce treatment intensity in head and neck cancer, as such patients have been shown to have a more favorable disease prognosis and response rate compared to patients with smoking-associated malignancies. However, despite phase II trials demonstrating favorable outcomes with reducing the dose of radiation [157,158], the phase II multi-center randomized NRG-HN005 (NCT 03952585) did not reach a non-inferiority threshold and has been closed temporarily [159]. In breast cancer, early data on the genomically adjusted radiation dose (GARD), a model combining gene-expression-based radiosensitivity index with the linear-quadratic model, may be promising for future efforts to individualize radiation dose [160,161]. Nonetheless, there is no doubt that future trials will leverage genetic information to better inform radiation dosing to optimize treatment outcomes and reduce the risks associated with dose-escalated radiation.

### 5.3. Volume Delineation 

One of the most challenging and unfortunately largely subjective roles of a radiation oncologist is to define the target volumes for RT. RT is individualized to each patient with target volumes delineated based on patients’ anatomy, disease status, and other characteristics. Radiogenomic data can be utilized to further inform practitioners throughout these clinical decisions. One consideration is the dose to normal tissue, particularly to organs at risk (OARs). For patients harboring radio-sensitizing mutations, it could be prudent to have more conservative constraints for OARs to minimize adverse effects, such as the volume of small bowel receiving a high dose (50–55 Gy), or moderate dose (30 Gy). These more stringent constraints might also necessitate the need to adjust target volume contours (e.g., reduction of elective nodal volume coverage), and in some cases, avoidance of radiation completely as noted above. In oropharyngeal squamous cell carcinoma (SCC), recent efforts, such as the EVADER phase II clinical trial (NCT 03822897), aim to evaluate the efficacy of biomarkers expression as a guide for target volume coverage for RT [162]. In the EVADER trial, patients with p16+ cancer are treated with volume-reduced elective nodal coverage and the primary objective is to evaluate the efficacy of this approach in impacting survival outcomes and toxicities. To the best of our knowledge, this is the only ongoing trial that assesses incorporating biomarkers as guidance for RT volume delineation. Future trials are needed, particularly focusing on germline mutations, to further “personalize” our treatment approach.

### 5.4. Combining with Systemic Therapy

Genetic information can be leveraged to guide the selection of concurrent systemic therapies with RT to further enhance the effect of radiation. Synthetic lethality is a term used to describe therapeutic agents that kill target cells by taking advantage of existing cellular defects [163]. Such a combined effect of the somatic mutation and targeted therapy can also sensitize tumor cells to sublethal doses of DNA damage, a concept known as synthetic cytotoxicity [163]. Genetic information can guide the selection of a known radiosensitizers to optimize the therapeutic ratio in favor of tumor killing without worsening toxicity. For example, inhibition of *PARP* has been shown to significantly increase the number of unrepaired DSBs in prostate cancer cells after irradiation [164].

In addition to targeted therapies, gene silencing can also be a potential strategy for potentiating the effect of RT. Genetic radiosensitizers work primarily by disrupting gene expressions through the use of short interfering RNAs (siRNAs) to further destabilize the genetic integrity of cancer cells to induce added DNA damage. An example of such a target is the *Nijmegen breakage syndrome-1 (NBS1)* gene which has an important role in HR repair of radiation-induced DSBs [165]. In non-small cell lung cancer cells, siRNA targeted for the *NBS1* gene has been shown to enhance radiosensitivity [165]. Another genetic target for radio-sensitization with promising effects is the *5-aminoimidazole-4-carboxamide ribonucleotide formyltransferase/IMP cyclohydrolase* (*ATIC)* gene, which encodes a protein in the de novo purine biosynthetic pathway [166]. *ATIC* inhibition via siRNA showed a significant increase in cell death following exposure to ionizing radiation. In cells not exposed to radiation, there was no detectable DNA damage, but cells were shifted into a more proliferative cell cycle phase, and thus a more radio-sensitized state [166]. These in vitro data have important implications for the potential role of genomically-targeted radiosensitizers. However, the translation of siRNA therapy remains limited due to several barriers to its development, such as chemical stability and glomerular filtration in blood circulation [167]. With improvements in technology, future studies are needed to focus on evaluating the clinical utility of combining these systemic therapies with RT

## 6. Precision Medicine in Today’s Radiation Oncology Clinic—The Example of Genomically-Guided Radiation Treatment in Breast Cancer 

Personalized medicine has a long and storied history in breast cancer medical oncology. Tamoxifen could be considered one of the field’s first truly personalized targeted therapies for patients with estrogen receptor-positive/human epidermal growth factor receptor-negative (ER+/HER2−) breast cancer. Tumor genomic assays, such as the 21-gene assay (commercially known as Oncotype DX^®^), currently play a significant role in adjuvant chemotherapy decisions for ER+/HER2− breast cancer [168]. For instance, breast cancer patients with germline BRCA1/2 mutations are candidates for a class of drugs called PARP inhibitors [169]. In addition, for patients with advanced ER+ breast cancer who have progressed on first-line endocrine therapy, the drug alpelisib, in combination with fulvestrant, is approved for those that have somatic mutations in PIK3CA kinase [170]. These are just a few of the many applications of precision medicine to breast cancer medical oncology.

On the other hand, the translation of personalizing RT has been slow. Currently, radiation oncologists use standard doses of radiation to eradicate microscopic disease in the adjuvant setting or to treat gross disease with or without concurrent chemotherapy. However, data are accumulating for the use of genomic assays and/or other biomarkers to help identify patients with breast cancer who may not benefit from adjuvant RT. For example, in the single-arm prospective LUMINA study post-lumpectomy patients with a low proliferative index (Ki67 ≤ 13.25%) in addition to patient and treatment-related factors (age ≥ 55 years; grade 1–2; tumor size ≤ 2 cm; axillary node-negative) had a 5-year risk of local-regional recurrence of 2.3% without adjuvant RT, suggesting that omission of RT may be appropriate in this low-risk patient population [171].

Randomized trials are ongoing to determine whether the use of tumor genomic assays may help identify patients who are candidates for omission of adjuvant whole breast radiation. One such trial is the NRG Oncology BR007 (DEBRA) study, in which patients aged 50–69 years old with ER+/HER2−, stage I (≤2 cm and node-negative) breast cancers with Oncotype recurrence score (RS) ≤18 treated with lumpectomy are randomized to radiation versus omission of radiation (NCT04852887). The EXPERT trial is taking a similar approach for patients aged ≥50 years old with stage I (node-negative), grade 1–2, HR+/HER2− breast cancers and uses the Prosigna (PAM50) assay to identify luminal A patients with low risk of recurrence to randomize patients to omission of radiation versus adjuvant RT (NCTC02889874). The Oncotype RS is also currently being used for patients with node-positive breast cancer as a potential biomarker to help determine which patients may not ^benefi17t^ from regional nodal irradiation (RNI). The MA39 study randomizes patients aged ≥ 40 years old with ER+/HER2− breast cancer with 1–3+ nodes after lumpectomy or 1–2+ nodes after mastectomy with Oncotype RS ≤ 25 to whole breast irradiation +/− RNI (after lumpectomy) or postmastectomy radiation (PMRT) versus no PMRT after mastectomy (NCT03488693). The study also includes patients with micrometastases and/or large tumors with negative axillary nodes (T3N0). As these studies complete accrual and the data matures, we anticipate that these genomic assays will potentially be used to personalize RT decisions in the next 10–15 years.

The assays described above were primarily developed to help guide adjuvant systemic therapy decisions in ER+/HER2− breast cancers. Several groups have also worked on developing radiation-specific genomic assays. These assays include the Radiation Sensitivity Index (RSI) [172], the Profile for the Omission of Local Adjuvant Radiation Treatment (POLAR) [173], and the Adjuvant Radiation Treatment Intensification Classifier (ARTIC) [174]. The ARTIC and POLAR assays are both prognostic and predictive, whereas the RSI is prognostic. For example, the POLAR assay is a 16-gene signature that was used to test and validate whether it could be used to identify patients with ER+/HER2−, node-negative breast cancers that may not benefit from adjuvant RT. Using data from two independent clinical trials, Sjostrom and colleagues demonstrated that patients defined as POLAR low-risk had low 10-year rates of local-regional recurrence (6–7%) without RT and that RT did not benefit these patients. However, patients identified as POLAR high-risk had higher rates of LRR and demonstrated a significant reduction in LRR risk with adjuvant RT from 19% to 8% (hazard ratio [HR] 0.43) in one of the trial cohort and from 22% to 8% (HR 0.25) in the second trial cohort [173]. While none of these radiation-specific assays are currently undergoing prospective validation in clinical trials, we anticipate that these studies will begin shortly and will add to our understanding of personalized RT.

In terms of personalizing radiation dose prescriptions for individual patients, the RSI and its related assay, the genomic-adjusted radiation dose (GARD), appear to hold the most promise. GARD scores were calculated based on the expression of ten pre-determined genes measured by the Affymetrix^®^ Gene Chip probe/gene array system (Affymetrix, Santa Clara, CA, USA) [160]. The GARD score has been shown to be an independent predictor for both relapse-free survival and distant metastasis-free survival in multiple tumor types [160]. There is significant interest in using the GARD score in patients with breast cancer, particularly the aggressive triple-negative (ER-/PR-/HER2−, TNBC) breast cancers. Rather than to help guide the omission of RT, the hope is that it can help identify patients who may require escalated doses of radiation for improved local-regional control. For example, Ahmed and colleagues have proposed an individualized dose range for patients with TNBC using the GARD assay [161]. We anticipate that future cooperative group clinical trials will incorporate the GARD or similar assays into trial design, particularly for TNBC.

## 7. Discussion and Future Direction

As summarized in Table 1, the current evidence supporting the role of radio-sensitizing genes is largely from in vitro/preclinical or observational studies. High-level scientific data, such as those from prospective randomized controlled trials (RCTs), validating the applicability of genetic results are lacking. In the near future, RCTs would need to be proposed to explore this topic.

At our institution, we currently have an ongoing non-interventional, prospective Precision Medicine Research Program. Patients who enroll have the option to undergo germline and somatic testing and are given an opportunity to meet with genetic counselors to discuss the test results. Observational studies such as these are necessary to gather the large-scale population-based data required to help inform individualized treatment responses based on genetic alterations. In the future, however, randomized and prospective clinical trials that incorporate germline data, as well as tumor-specific genetic and radiomic features, are necessary to help us better understand if the utilization of genetic data can yield clinically meaningful improvements in cancer control and toxicity outcomes (Figure 3).

Genetic testing results have great potential to further inform RT planning and recommendations to improve clinical outcomes. To move the needle forward, ongoing efforts are required at multiple levels and disciplines. Efforts are needed at the patient level to validate the significance of individual genetic testing results as well as at the population level, to demonstrate the role of genetic testing in increasing the effectiveness of RT, ideally from six randomized controlled trials.

There is not a single ideal trial design to evaluate the clinical application of radio-genomics, each with its pros and cons [175]. Therefore, the ideal trial design would be dependent on the research question. For example, one theoretical trial design to study an altered course of RT would be to randomize patients in a 2 × 2 fashion: first, based on the presence of tumor and/or germline mutations of radio-sensitizing genes (or a particular molecular signature), and then randomized into two intervention groups with one arm receiving an altered course of radiation and the control arm receiving the standard of care. Outcomes to monitor may include rates of local recurrence, survival, and side-effects. However, concern remains in interpreting the results of genetic testing, as the majority of variants identified in radiosensitizing genes are those of uncertain significance.

Nevertheless, the results from these trials will provide us with critical insights into the potential impact of genetic testing on cancer care. In addition to clinical outcomes, data from these trials can also be used for the development of predictive models to assess the risks associated with radiation. As such, continued studies are imperative to unravel the intricate roles of complex mutations in radiation response and develop more reliable, broadly applicable multi-gene predictive models that can aid clinicians in making informed RT treatment decisions for individual patients.

## 8. Conclusions

In this review, we provide an update on the current evidence on “radio-sensitizing genes”. The role of genetics in radiosensitivity is complex; our current understanding of the many implicated pathways and their compensatory mechanisms is only the “tip of the iceberg”. The number of mutations identified to have an association with radiosensitivity will only continue to increase over time. More importantly, future endeavors would need to focus on establishing the clinical impact of these mutations and whether they individuals or a combination thereof should be utilized in guiding treatment decisions for RT.

## Figures and Tables

**Figure 1 cancers-15-05314-f001:**
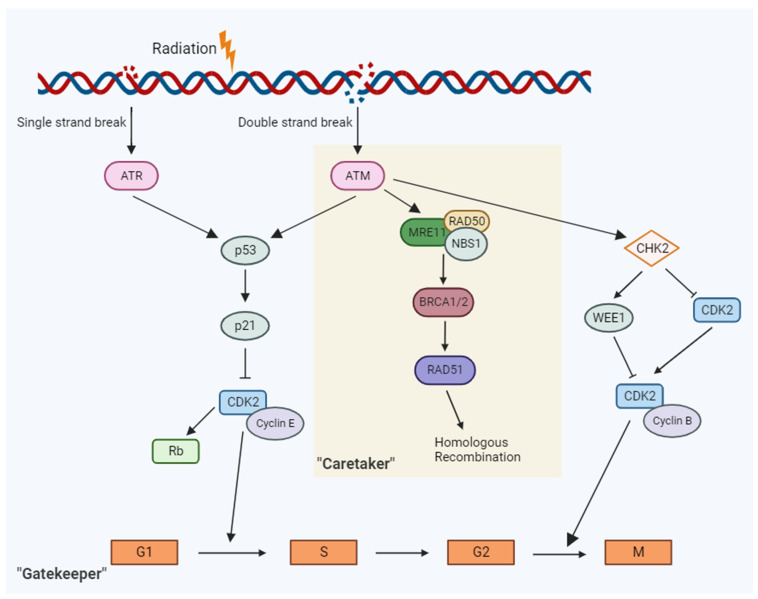
In response to DNA damage response from radiation, a cascade of events along the “gatekeep” and “caretaker” pathways are activated that ultimately leads to cell cycle arrest or DNA repair.

**Figure 2 cancers-15-05314-f002:**
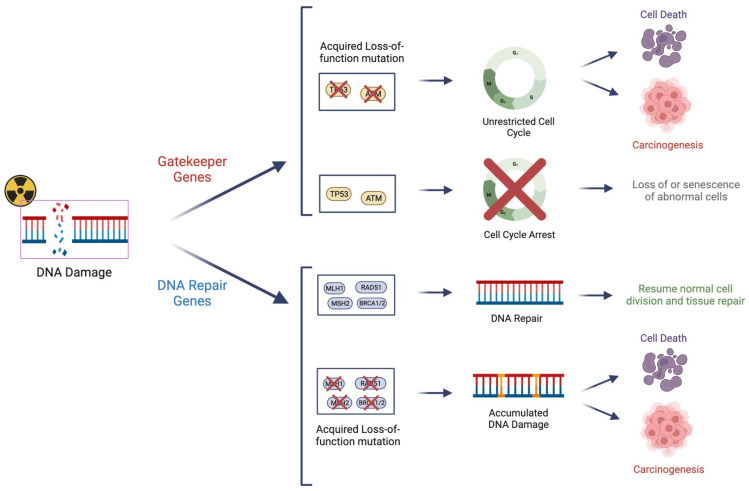
Cell cycle checkpoint “gatekeeper” genes and DNA repair “caretaker” genes are important regulators in ensuring the integrity of the genome prior to cell division. Mutations of these genes can lead to their loss of function and dysfunction of the regulatory mechanism, as well as potential uncontrolled cell growth “carcinogenesis”. These genes also have important roles in conferring “radio-sensitivity” due to impaired surveillance and/or repair of radiation-induced DNA damage.

**Figure 3 cancers-15-05314-f003:**
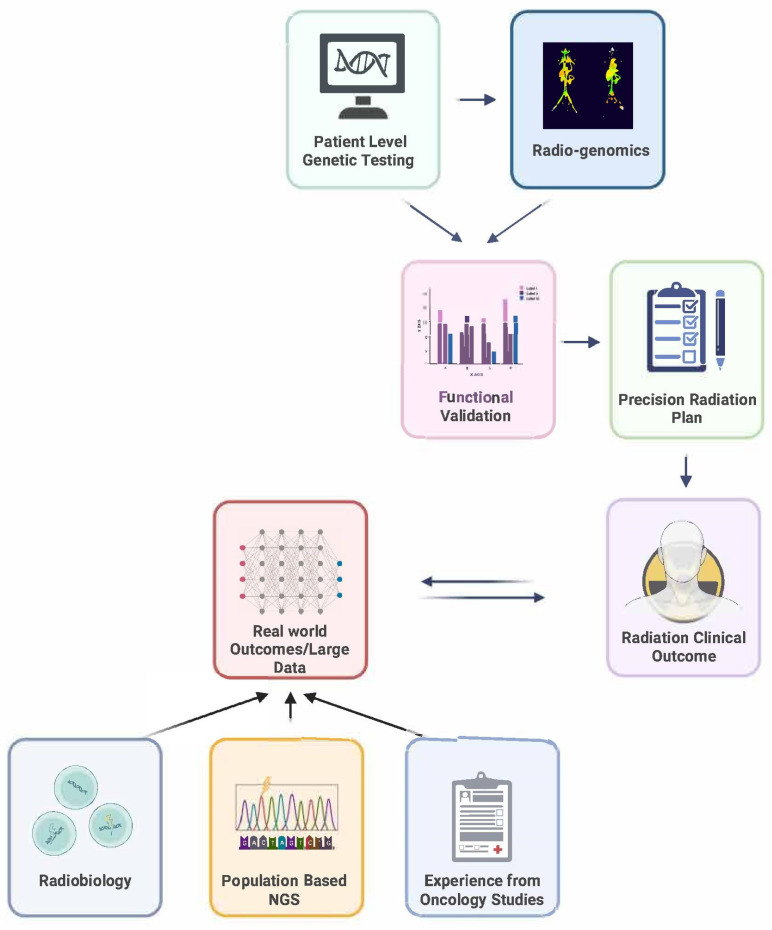
The Paradigm of Genomically-Guided Radiation Treatment.

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
