# Peer review of "Moving the Needle Forward in Genomically-Guided Precision Radiation Treatment"

_cancers, 2023, doi:10.3390/cancers15225314_

Round 1

Reviewer 1 Report

This review is titled “Moving the Needle Forward in Genomic-Guided Precision 2 Radiotherapy.”

It has two main objectives: 1) describing genes, whose mutations are known to have a radio-induced toxicity effect, and 2) describing the implications on the treatments personalization.

These objectives are only partially achieved. In fact, although the reading is clear with good English level and the arguments structure is fine, the review is long and rich in circumstantial arguments, in some parts poor in contents and in others off topic with respect to manuscript topic.

In fact, the reader would expect a real update of genomic studies, conducted on large cohorts of patients, aiming of identifying genes and their mutations with repercussions of increased toxicity after radiotherapy. Instead, in the section 2 and 3, the authors only describe the main genes involved in radiotoxicity, already known for some time. Thus, this section seem more a book chapter, which focus on this topic with a didactic approach. The review should be enriched by referring to large genomic studies, such as those cited in the review Forte et al. Transl Cancer Res 2017;6(Suppl 5):S852-S874, describing very large studies including hundreds of patients (Rapper, RADIOGEN, GenePARE, CCI etc) and citing associations such as the Radiogenomics Consortium (RGC) https://epi.grants.cancer.gov/radiogenomics/ . I believe these are important lacking information in a review that would be updated and informative.

The second part of the review concerns the implications that these mutations have on the therapeutic plans of patients.

The section 5.2, regarding dose customization, should include the example regarding the GARD score, reported on page 18, which would indeed imply a re-evaluation of the prescribed dose based on the outcome of the GARD test. On the other hand, in section 5 it seems useless to talk about customization of the volume to be irradiated (GTV), as the examples shown do not report inclusion criteria attributable to genetic mutations. Similarly, paragraph 5.4 seems useless, since it is discussed the impact of combined therapies with molecules targeting some key genes involved in radiotoxicity, without reporting examples in which these genes were muted/altered. An argument that has nothing to do with a defined paragraph: "Implications of Genomics in Radiation Therapy Clinical Decision Making", in which the reader would expect to read examples of studies concerning patients with alterations on sensitive genes, treated with combined therapies. Also, they cite in vitro studies, which easily lead to a radiosensitizing results, even when the combination should occurs with a non-targeted molecule. Therefore, in the context of paragraph 5, section 5.4 should be deleted because it is off topic.

Instead, the paragraph 6 in the third session is useful and finally center the review subject with fitting examples.

In summary, the following major revisions are proposed:

·    Section 3.2: Revise the paragraph on MMR genes, as line 263-266 seems to restart the introduction on these genes.

·      Section 5:2 Include here the GARD assay example reported at page 18.

·      Section 5.3 Delete or rewrite with relevant examples inherent the review subject;

·       Section 5.4 Deleting text and tables

·   Section 6. Implement this section of other studies with large patient cohorts, as explained above.

Overall, reduce the review text to main concepts, deleting circumstantial arguments.

Best Regards

Reviewer 2 Report

In this review, the authors explore the concept of genomically-guided precision radiotherapy, but I have several concerns with the content:

  1. Unclear Aim: The purpose of summarizing all available literature is not well-defined. It's essential to clarify why the authors deemed it necessary to undertake this task at this particular time. Perhaps waiting for more novel works to emerge could be considered.
  2. Missing Reference: In the Introduction, the first paragraph lacks a proper reference (L44-53).
  3. Ambiguity in Abbreviation: Please clarify whether "RT" in L55 refers to radiation treatment or radiotherapy, as previously mentioned in L24.
  4. Use of Italics: The reason behind using Italic font for "ATM" in L57 should be explained.
  5. Table Order: It would be more appropriate to present Table 1 in chronological order (by year) to improve readability.
  6. Expanding Section 4: The authors should consider expanding the content in Section 4 to include a broader range of molecular testing types and provide a more comprehensive review.
  7. Chronological Order for Tables: Tables 2 and 3 should also be arranged in chronological order to facilitate understanding.
  8. Diverse Examples: Section 6 would benefit from including examples beyond breast cancer. Considering other cancers, such as lung, melanoma, ovarian, thyroid, and colorectal cancer, would enhance the review's applicability.

I have no comment on the quality of English in this submission.

Reviewer 3 Report

In the review by Tam and colleagues, the authors provide an overview of the relationship between genetic alterations and radiosensitizing effects of both tumor and normal tissue in the context of radiotherapy. Radiosensitizing effects due to genetic germline of somatic mutations are well known, particularly for DNA repair deficiency syndromes. However, low-penetrance genetic alterations that may influence the radiation response and risk of the general population remain largely unconsidered for the use of ionizing radiation in routine clinical practice. The authors, therefore, address an interesting point that is relevant to radiosensitization of tumors as well as medical radiation protection. The article is relatively broad, so many topics are not covered in depth. It provides a rudimentary overview and a rather solid base knowledge for clinicians.

From my point of view, some issues should be considered in a major revision before considering publication:

1)      Tabelle 1: What is this exemplary selection of studies based on? There is a plethora of work on this topic. Several very relevant gene defects, which also play a role in radiosensitizing tumors and normal tissue are not considered, e.g., LIGIV, NBS1, ATR (Seckel), Fanconi, Bloom, PARP1.... In contrast, for ATM, although highly prominent, 4 clinical studies are cited.

2)      In Table 1 the authors also cite in vitro studies using functional bioassays to monitor DNA damage induction and repair. How do the authors judge the role of functional bioassays versus genetic testing in the clinical setting?  A (brief) comparison would be of interest to the reader.

3)      An illustration providing an overview of the known major players of the DNA damage response (DNA repair, cell cycle checkpoints) and their radiosensitizing effects by dysfunctional genes and their products through genetic or epigenetic mutations would be helpful.

4)      Line 107: Why is hypoxia mentioned here, which plays a role mainly or only in solid tumors? Since the authors are talking about genetic predispositions, only intrinsic factors should be noted. Or are there, e.g., genetic factors (HIF1 ?) that have radioprotective effects through hypoxia in normal tissue?

5)      Line 177: What about the G2/M checkpoint and the involvement of kinases such as CHK1 or Wee1, which are also considered in radiosensitization strategies? Is there evidence of a genetic basis for the impairment of the G2/M arrest, which may be a more potent radiosensitizer than the G1/S arrest due to mitotic catastrophes. Notably, G2/M arrest also plays an important role in p53-deficient tumor cells with a dysfunctional G1/S checkpoint. Why are the authors limited to G1/S arrest?

6)      Line 236: Why is the role of ATM in the activation of p53 not discussed in the previous section (Gatekeepers)? There, p53 and ATM are mentioned in line 183, but ATM is not addressed in the Caretaker section but Rb.

7)      Lines 247-272: The section on MMR and Lynch syndrome should be revised. There is a lot of repetition and redundancy here, especially in lines 262-272. The evidence for MMR concerning radiosensitization is also rather limited.

8)      Line 272: For Rad51, the important involvement in homologous recombination should be mentioned. HR, for example, is not mentioned until line 288 for BRCA1/2. In general, more attention should be paid to DNA repair pathways at the beginning of section 3.2. They should be differentiated better and the DNA repair proteins should always be assigned to the respective repair pathways.

9)      Line 304: However, the application of functional DNA repair assays indicates limited repair of radiation-induced DSBs by HR in fibroblasts from patients with BRCA2 but not BRCA1 heterozygous germline mutations: https://doi.org/10.1016/j.radonc.2011.05.043. This study is also an example of the large number of publications not included in Table 1 and in general. Also, a discrepancy between the results of functional DNA repair assays and clinical radiosensitivity should be discussed, as often observed in Fanconi anemia, for example (https://doi.org/10.1016/j.dnarep.2020.102992). Herein lies an advantage of genetic testing compared to functional bioassays before the start of radiation therapy.

10)   Line 330: FISH is a cytogenetic method of chromosome analysis, not protein analysis. Instead of or in addition to citing the detection of HER2, perhaps the authors should briefly refer to the application of such methodologies to the topic at hand: https://doi.org/10.1016/j.radonc.2019.06.038

11)   Line 439: How does genetic targeting compare to molecular targeting with pharmaceutical inhibitors? How can we achieve targeted inactivation of genes in the tumor? What is the difference here? Targeted silencing of genes in the tumor would not represent a systemic but local therapy.

12)   Table 3: For Chen et al. 2018, EGFR double mutations in NSCLC are not an overexpression nor was an inhibitor applied. Perhaps it is HRDnes, i.e. downregulation of HR activity by downregulation of EGFR signaling.

13)   Line 497: As noted in comment 1, PARP should be introduced earlier. It also represents a potential radiosensitizer, e.g. https://doi.org/10.1002/ijc.32018. Concerning comment 8), this study also shows that other repair pathways can play a role, especially in tumor cells, such as alt-NHEJ/MMEJ and e.g. POLQ is also relevant here, e.g. https://doi.org/10.1038/nature14184. Also, the authors should mention the concept of synthetic lethality or synthetic cytotoxicity as a strategy for radiosensitization.

Round 2

Reviewer 1 Report

The authors have significantly updated the original version of the manuscript, taking into consideration my raccomandation. 

I only have few further suggestions as minor revisions. 

1)    The authors should rewrite the abstract, as they focused it only on the role of genetic mutations for their predictive use of radiosensitivity. Instead, they also have to refer to the genomically-guided studies, described in the second part of the manuscript, which are the only ones really introduced in the clinical practise.

2)    Please, remove the improper use of the term “preclinical” studies, when it is not referred to animal studies, rather use “in vitro” studies if they use cells.

3)    The 5.5 sections only report generic considerations. Thus, it should be deleted. Instead, the next section 6 should include in the title the concept that the described studies are “genomically-guided”. Thus, this section should be titled more properly “Precision Medicine in Today’s Radiation Oncology Clinic – The example of Genomically-Guided Radiotherapy in Breast Cancer”

After these minor revisions processing, I will have no other request and I will suggest this review for publication on Cancer journal.  

Regards, 

GIF

Reviewer 2 Report

I am satisfied with the modifications and additional contents made by the authors as per my comments. The overall presentation and quality of the manuscript are improved.

Author Response

We would like to thank the reviewer once more for their thoughtful comment.

Reviewer 3 Report

The authors have edited all comments completely and satisfactorily. Only the numbering of the figures needs to be updated (Figures 2 and 3).

I therefore recommend publication in the present version and congratulate the authors on their thorough work.
